# Effects of Different Primary Processing Methods on the Flavor of *Coffea arabica* Beans by Metabolomics

Xiaojing Shen [1,2], Chengting Zi [1,2], Yuanjun Yang [1], Qi Wang [1,3], Zhenlai Zhang [1], Junwen Shao [1], Pincai Zhao [1], Kunyi Liu [3,*], Xingyu Li [1,*] and Jiangping Fan [1,*]

[1] College of Science, College of Food Science and Technology, College of Water Resources and Hydraulic Engineering, Yunnan Agricultural University, Kunming 650201, China; mumu202107@163.com (Q.W.)

[2] Research Center for Agricultural Chemistry Yunnan Agricultural University, Yunnan Agricultural University, Kunming 650201, China

[3] School of Wuliangye Technology and Food Engineering, Yibin Vocational and Technical College, Yibin 644003, China

[*] Correspondence: ben.91@163.com (K.L.); lixingyu@ynau.edu.cn (X.L.); 1993033@ynau.edu.cn (J.F.)

**Abstract:** The primary processing method of coffee plays a crucial role in determining its flavor profile. In this study, roasted coffee beans were subjected to three primary processing methods, i.e., natural processing (SC), washed processing (WC), and honey processing (MC), that were analyzed by LC-MS/MS and GC-MS metabolomics. Additionally, sensory evaluation was conducted by the Specialty Coffee Association of America (SCAA) to assess coffee flavor characteristics. The results showed that 2642 non-volatile compounds and 176 volatile compounds were detected across the three primary processing methods. Furthermore, significant differentially changed non-volatile compounds (DCnVCs) and volatile compounds (DCVCs) were detected among SC/WC (137 non-volatile compounds; 32 volatile compounds), MC/SC (103 non-volatile compounds; 25 volatile compounds), and MC/WC (20 non-volatile compounds; 9 volatile compounds). Notable compounds, such as lichenin, 6-gingerdiol 5-acetate, 3-fluoro-2-hydroxyquinoline, and 4-(4-butyl-2,5-dioxo-3-methyl-3-phenyl-1-pyrrolidiny)benzenesulfonamide, were identified as important DCnVCs, while ethyl alpha-D-glucopyranoside, 2,3-butanediol, maltol, and pentane-1,2,5-triol were identified as significant DCVCs in SC/WC. In MC/SC, 3-fluoro-2-hydroxyquinoline, etimicin, lichenin, and imazamox were important DCnVCs, whereas ethyl alpha-D-glucopyranoside, 2-pyrrolidinone, furfuryl alcohol, and pentane-1,2,5-triol were import DCVCs. Lastly, MC/WC samples exhibited notable DCnVCS, such as (S)-2-hydroxy-2-phenylacetonitrile O-[b-D-apiosyl-1->2]-b-D-glucoside], CMP-2-aminoethyphosphonate, talipexole, and neoconvallatoxoloside, along with DCVCS including citric acid, mannonic acid, gamma-lactone, 3-(1-hydroxy-1-methylethyl)benzonitrile, and maltol. Therefore, the primary processing method was a useful influence factor for coffee compositions.

**Keywords:** primary processing; *Coffea arabica*; roasted coffee beans; coffee flavor; non-volatile compound; volatile compound

## 1. Introduction

Coffee is the second most important commodity traded in the world, with an estimated production of 6.6 million bags of coffee forecasted for 2022/23 by the USDA (United States Department of Agriculture Foreign Agricultural Service). Global coffee consumption is expected to reach 167.9 million, and the ICO Composite Indicator Price (I-CIP) reported an average price of 157.19 US cents/lb for coffee beans in December 2022. In particular, China contributes to global coffee production, and its primary coffee plantation area is located in the Yunnan province, which accounts for over 95% of coffee plants in the country.

The increasing popularity of coffee is attributed to its rich and compelling flavors, leading to significant growth in the coffee market in recent years [1]. From post-harvest to the cup, various factors impact coffee aroma and taste, including genetics, shade, elevation,

brewing, and serving methods, and most importantly, coffee post-harvest processing, which plays a crucial role in modulating coffee aroma and coffee bean quality [2–4]. Coffee flavor precursors, such as carbohydrates, proteins, lipids, amino acids, organic acids, alkaloids, and phenolic compounds, are present in green coffee beans and undergo intricate reactions during coffee roasting stages [5,6], which lead to the formation of over 1000 volatile compounds in coffee, significantly influencing its overall quality as a beverage [7].

The first stage in coffee production involves post-harvest processing, which is necessary for obtaining green coffee beans. Three common processing methods are employed at this stage: (i) dry processing or natural processing; (ii) wet processing or washed processing; and (iii) semi-dry processing or the honey method [1,8]. Dry processing combines fermentation and drying, with whole pericarp berries left to dry naturally or artificially for approximately 10–25 days, thus resulting in leathery fruits covered by the pericarp, and then dried cherries are peeled to reveal green coffee beans [1,8]. Wet processing employs coffee cherries de-pulped to eliminate the exocarp, and then they are subjected to submerged fermentation for 12–36 h. After fermentation, coffee beans are washed, dried for 5–10 days, and peeled to obtain green coffee beans [1,8]. The semi-dry method is a hybrid of the wet and dry processing methods, wherein berries are de-pulped and dried while beans are still partially covered by mucilage [8].

Each processing method imparts distinct aromas and flavors to coffee products and beverages [9]. For instance, dry processing tends to result in coffee with low acidity, exotic flavors, and more body flavor [10]. In contrast, washed processing generally results in cleaner, lighter, slightly fruity characteristics, with a light and soft body and a higher level of acidity [10], which often yields a high-quality coffee.

In this study, we analyzed the metabolites and conducted a cupping analysis of roasted coffee beans obtained via the three processing methods, namely dry processing (SC), wet processing (WC), and semi-dry processing (MC), to investigate the impact of primary processing on coffee flavor.

## 2. Materials and Methods

### 2.1. Materials and Chemical Standards

The raw material used for roasted coffee beans in this experiment was *Coffea arabica* obtained at Pu-er City, Yunnan Province, China. After harvesting, coffee cherries were processed to obtain green coffee beans using three processing methods: natural processing (SC), washed processing (WC), and honey processing (MC). Then, green coffee beans underwent medium roasting to obtain roasted coffee beans used for subsequent analysis. Methyl alcohol of high-performance liquid chromatography (HPLC) grade, acetonitrile, and propyl alcohol were purchased from Fisher Co., Ltd. (Shanghai, China).

### 2.2. Analysis of Non-Volatile Compounds

Metabolites in roasted coffee beans were extracted and analyzed using a liquid chromatography-mass spectrometry (LC-MS/MS) based metabolomics approach performed by Majorbio Co. Ltd., Shanghai, China. Roasted coffee powder (RCP) samples (50 mg) were accurately weighed and extracted using 0.4 mL 80% methanol solution with 0.02 mg/mL L-2-chlorophenylalanin as the internal standard. The samples were then centrifuged at $13,000 \times g$ for 15 min at 4 °C, and the supernatant was transferred to sample vials for LC-MS/MS analysis. Quality control (QC) samples were prepared by combining equal volumes of all samples to monitor analysis stability.

Samples were injected into a UHPLC-Q-Exactive system from Thermo Fisher Scientific for LC-MS analysis [11]. The chromatographic separation was performed using an HSS T3 C18 column (2.1 × 100 mm, 1.8 μm; Waters Corporation, Milford, MA, USA) at 40 °C, under the following LC parameters: injection volume, 2 μL; and flow rate, 0.4 mL/min. The mobile phase consisted of a mixture of (A) 0.1% formic acid in water: acetonitrile (95:5, *v/v*) and (B) 0.1% formic acid in acetonitrile: isopropanol: water (47.5:47.5:5, *v/v*). The gradient elution was as follows: 0–5% B for 0–0.1 min; 5–25% B for 0.1–2 min; 25–100% B for 2–9 min; 100%

B for 9–13 min; and 100–0% B for 13–13.1 min; then 0% B for 13.1–16 min to equilibrating the systems. The effluent was alternatively connected to an electrospray ionization (ESI) operating source, and the optimal conditions were: heater temperature, 400 °C; sheath gas flow rate, 40 arb; aux gas flow rate, 10 arb; ion-spray voltage floating (ISVF), $-2800$ V in negative mode and 3500 V in positive mode; and normalized collision energy, 20–40–60 V for MS/MS. The full MS resolution was set at 70,000, and the MS/MS resolution was 17,500. The detection range covered a mass range of 70–1050 $m/z$. The LC-MS was preprocessed using Progenesis QI software (Waters Corporation, USA). Simultaneously, metabolites were searched and identified by the HMDB Metlin and Majorbio Database. The response intensity of the sample mass spectrum peaks was normalized using the sum normalization method, and variables with a relative standard deviation (RSD) > 30% of QC samples were removed, followed by log10 calculations.

### 2.3. Analysis of Volatile Compounds

Volatile compounds in roasted coffee beans were evaluated using a gas chromatography-mass spectrometry (GC-MS)-based metabolomics approach performed by Majorbio Co. Ltd., Shanghai, China. RCP (50 mg) was accurately weighed and subjected to extraction using methanol: water (80:20, *v/v*) through an ultrasound method for 30 min. Following extraction, oximation, and derivatization, reactions were performed within 90 and 60 min. A model 8890B GC instrument and a 5977B mass spectrometer (Agilent, Santa Clara, CA, USA) were used for GC-MS analysis. A DB-5MS (40 m × 0.25 mm × 0.25 μm) capillary column was used for the identification and quantification of volatile compounds. The carrier gas used was 99.999% helium with a 1.0 mL/min column flow. The column temperature program was set to 60 °C, held for 0.5 min, and then raised to 310 °C at a rate of 8 °C/min. The transfer line, ion source, and quadrupole mass detector temperatures were set to 310 °C, 280 °C, and 150 °C, respectively. Mass spectra were recorded in the electron impact (EI) ionization mode at 70 eV and scanned in the $m/z$ range of 50–500.

GC-MS data were preprocessed using MassHunter Workstation quantitative analysis (v10.0.707.0) software [12]. The metabolites were identified and searched using the Fiehn and NIST public databases, and the data were uploaded to the Majorbio cloud platform for analysis. The response intensity of the sample mass spectrum peaks was normalized using the sum normalization method, and variables with an RSD > 30% of QC samples were removed, followed by log10 calculations.

### 2.4. Sensory Evaluation

Cupping analysis was conducted following the SCAA cupping protocol (Specialty Coffee Association of America, 2015) by certified professionals with expertise in cupping analysis. Coffee beans used for cupping tests were subjected to medium roasting. Ten attributes were evaluated: fragrance/aroma, flavor, aftertaste, acidity, body, balance, overall impression, uniformity, sweetness, and clean cup. Uniformity, sweetness, and clean cup were classified under the "objective" category, assessing the absence of defects. Other attributes were categorized as "subjective" and were scored based on their quality on a scale of 6 to 10 points in intervals of 0.25 points. Additionally, tasters described the characteristic flavors of each sample.

### 2.5. Statistical Analysis

Statistical analyses were performed using IBM SPSS Statistics 26.0 (SPSS Inc., Chicago, IL, USA). All results from three replicates were presented as the mean value $\pm$ standard deviation (SD). Variable importance in projection (VIP) analysis ranked the overall contribution of each variable to the OPLS-DA model. Those variables with VIP > 1.0, $p < 0.05$, and fold change (FC) > 2 or <0.5 were classified as differentially changed non-volatile compounds (DCnVCs) or differentially changed volatile compounds (DCVCs) [13].

## 3. Results and Discussion

### 3.1. Analysis of Differentially Changed Non-Volatile Compounds (DCnVCs)

Metabolomic is an essential analytical method for coffee studies, allowing for coffee varietal classification [14], the elucidation of the formation mechanism of coffee flavor precursors [15], the determination of the influence of extraction method on coffee flavor [16], and coffee by-products research [17], among others. Coffee is known for its rich composition, which includes alkaloids, benzene and its derivatives, phenylpropanoids, amino acids and their derivatives, lipids, heterocyclic compounds, carboxylic acids and their derivatives, and saccharides [18]. Untargeted LC/MS profiling has successfully identified coffee flavor compounds [19], and non-volatile compounds (nVCs) have been found to be related to coffee flavor. In this study, 2642 nVCs were detected in coffee samples processed using different methods. These were classified into 17 super-classes, as shown in Figure 1. The super-class with the highest number of compounds was lipids and lipid-like molecules (527 nVCs), followed by organoheterocyclic compounds (495 nVCs), organic acids and their derivatives (479 nVCs), organic oxygen compounds (365 nVCs), phenylpropanoids and polyketides (306 nVCs), benzenoids (258 nVCs), nucleosides, nucleotides, and analogs (73 nVCs), alkaloids and their derivatives (38 nVCs), organic nitrogen compounds (32 nVCs), hydrocarbons (9 nVCs), hydrocarbon derivatives (4 nVCs), lignans, neolignans, and related compounds (4 nVCs), organic 1,3-dipolar compounds (2 nVCs), homogeneous non-metal compounds (2 nVCs), organohalogen compounds (1 nVCs), acetylides (1 nVCs), and others (46 nVCs).

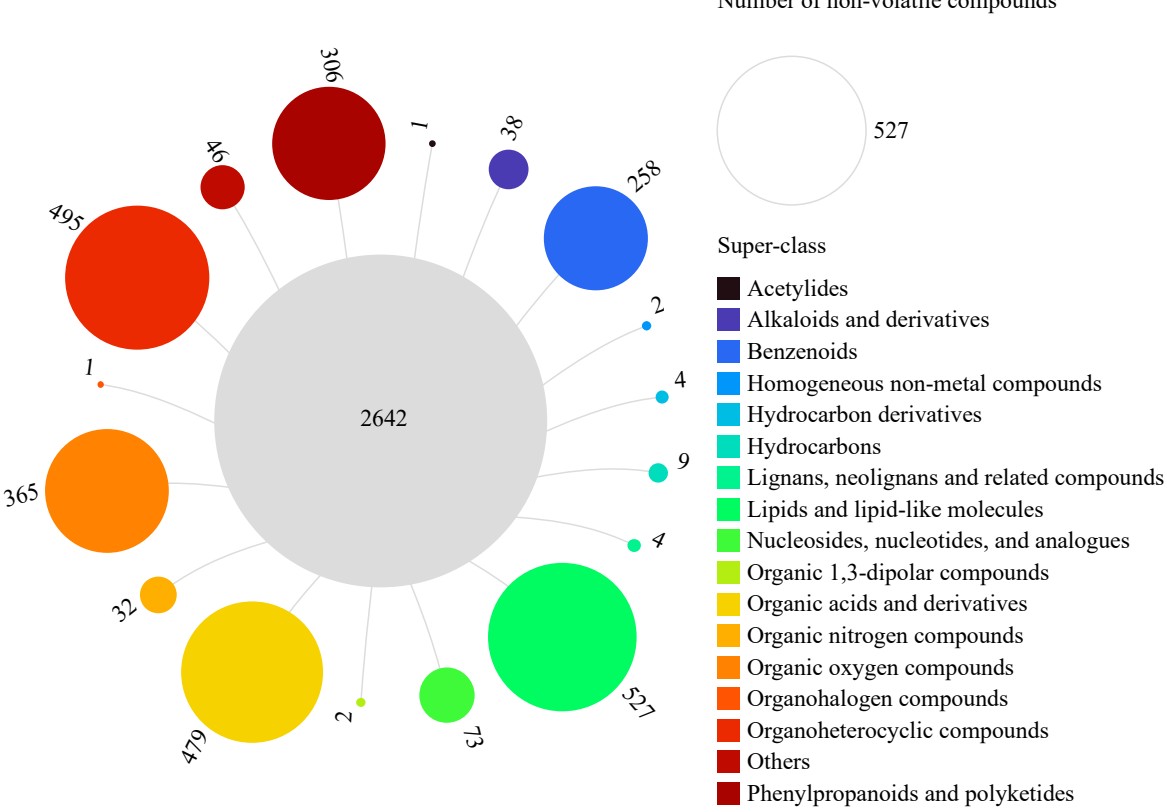

**Figure 1.** Super-classes of non-volatile compounds from different coffee primary processing methods.

Further grouping of these nVCs resulted in 163 classes, with carboxylic acids and derivatives being the largest class (398 nVCs); followed by organooxygen compounds (364 nVCs); fatty acyls (209 nVCs); prenol lipids (158 nVCs); benzene and substituted derivatives (138 nVCs); flavonoids (91 nVCs); steroids and steroid derivatives (81 nVCs); coumarins and derivatives (66 nVCs); glycerophospholipids (65 nVCs); phenols (63 nVCs); indoles and derivatives (61 nVCs); pyridines and derivatives (52 nVCs); cinnamic acids and

derivatives (51 nVCs); benzopyrans (40 nVCs); organonitrogen compounds (32 nVCs); keto acids and derivatives (29 nVCs); imidazopyrimidines (28 nVCs); quinolines and derivatives (27 nVCs); isoflavonoids (26 nVCs); lactones (23 nVCs); naphthalenes (22 nVCs); purine nucleosides (22 nVCs); hydroxy acids and derivatives (21 nVCs); pyrimidine nucleosides (21 nVCs); phenylpropanoic acids (19 nVCs); dihydrofurans (18 nVCs); diazines (18 nVCs); pyrans (17 nVCs); peptidomimetics (16 nVCs); piperidines (13 nVCs); heteroaromatic compounds (12 nVCs); azoles (12 nVCs); macrolides and analogues (11 non-volatile compounds); phenol ethers (11 nVCs); pyrrolidines (11 nVCs); pteridines and derivatives (10 nVCs); and benzodioxoles (10 nVCs); among others.

nVCs, such as alkaloids, lipids, chlorogenic acid, and carbohydrates, are important to the sensory quality of coffee [20]. For instance, alkaloids contribute to the bitter flavor of coffee [20]. Caffeine, an alkaloid compound found in coffee, influences the perceived strength, body, and bitterness of coffee, while trigonelline contributes to the overall aroma [20]. Coffee lipids were crucial in the coffee brew, encompassing different classes and accounting for about 17% of dry bean weight [21]. They form the crema emulsion of espresso coffee, carrying flavor volatiles and fat-soluble vitamins that contribute to the perceived texture and mouthfeel of the coffee brew [20]. Additionally, coffee lipids were discriminant markers for differentiating coffee species, roasts, and maturation levels. For instance, TG 48:2, TG 52:5;1O, C22(OH)− 5HT, and DG 36:0 have been used to differentiate *C. aarabica* and *C. robusta* [21–23]. In addition, chlorogenic acids can contribute to the astringency and bitterness of coffee [19]. Moreover, chlorogenic lactones, the reaction products of chlorogenic acids and quinic acid, can increase bitterness [20]. For instance, 3-*O*-caffeoyl-γ-quinide and 4-*O*-caffeoyl-γ-quinide could impact coffee flavor stability [24]. 3-*O*-caffeoyl-4-*O*-3-methylbutanoylquinic acid and 3-*O*-caffeoyl-4-*O*-3-methylbutanoyl-1,5-quinide have been found to significantly increase flavor, aroma, aftertaste, acidity, body, balance, and overall impression attribute scores [19]. Furthermore, 3-*O*-caffeoyl-4-*O*-3-methylbutanoylquinic acid serves as a chemical marker for green bean quality [19].

In this study, nine caffeoylquinic acids, including four mono-caffeoylquinic acids (cis-5-caffeoylquinic acid, 1-caffeoylquinic acid, 1-*O*-caffeoylquinic acid, and 5-caffeoylquinic acid), four di-caffeoylquinic acids (dicaffeoylquinic acid, 1,3-dicaffeoylquinic acid, 3,4-di-*O*-caffeoylquinic acid, and 1,5-dicaffeoylquinic acid), one tri-caffeoylquinic acid (3,4,5-tricaffeoylquinic acid), and five feruloylquinic acids, including three feruloylquinic acid (feruloylquinic acid, 3-feruloylquinic acid, and 3-*O*-feruloylquinic acid) and two caffeoyl-feruloylquinic acids (3-caffeoyl-4-feruloylquinic acid and 4-*O*-caffeoyl-3-*O*-feruloylquinic acid), were identified in different primary processing methods. Interestingly, 1-*O*-caffeoylquinic acid, previously confirmed in *C. arabica* by Asamenew et al. [25], exhibited a VIP value greater than 1 and was also detected in *C. arabica* from the Yunnan province.

To gain an overview of the DCnVCs between SC, WC, and MC, a comparison was made between the processing methods. As shown in Figure 2A, in the SC to WC comparison, the relative levels of 39 nVCs were decreased significantly (VIP > 1.0, $p < 0.05$, and FC < 0.5), whereas the relative levels of 98 nVCs in SC/WC were increased significantly (VIP > 1.0, $p < 0.05$, and FC > 2). These 137 DCnVCs were related to various compounds, including lipids and lipid-like molecules (34 nVCs, e.g., 5-acetamidovalerate, dihydrocarvone, dehydroepiandrosterone, etc.), organic acids and their derivatives (30 nVCs, e.g., 4-hydroxystachydrine, (3S)-3,6-diaminohexanoate, etc.), organoheterocyclic compounds (23 nVCs, e.g., pinolidoxin, gluconolactone, theophylline, etc.), organic oxygen compounds (22 nVCs, e.g., 1-*O*-caffeoylquinic acid, L-lyxonic acid, D-gluconic acid, pantothenol, inulobiose, etc.), phenylpropanoids and polyketides (7 nVCs, e.g., liquiritin, zeranol, calceolarioside E, etc.), benzenoids (7 nVCs, e.g., dopamine, 4-ethoxybenzamide, salutaridinol, etc.), organic nitrogen compounds (4 nVCs, bemcentinib, agmatine, dimethylethanolamine, and N,N-dimethylsphingosine), alkaloids and their derivatives (3 nVCs, isoguvacine, harmane, and pilocarpine), nucleosides, nucleotides, and analogs (2 nVCs, 3-methyluridine and xanthosine), and others (5 nVCs).

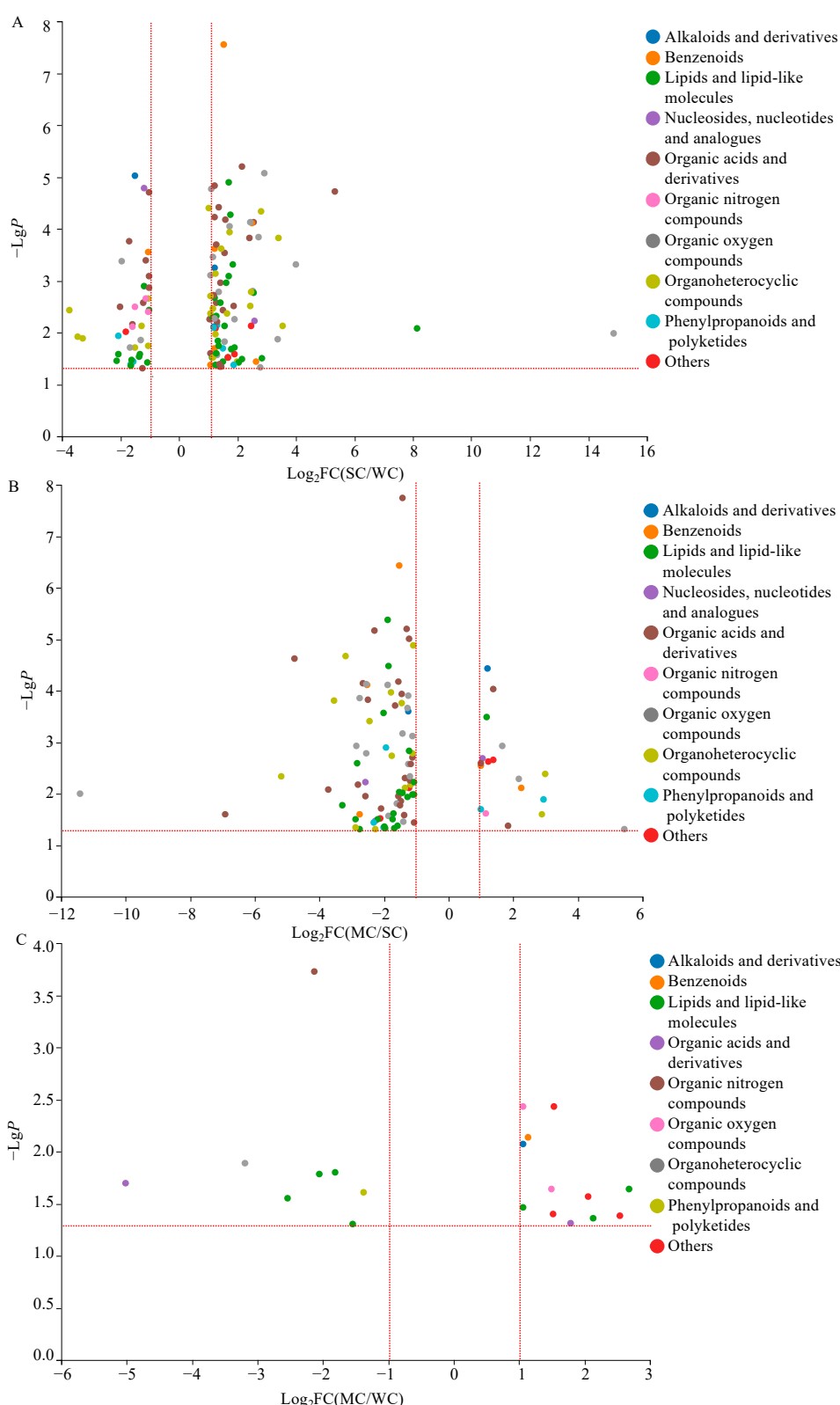

**Figure 2.** Differentially changed non-volatile compounds (DCnVCs) between natural processing (SC), washed processing (WC), and honey processing (MC), respectively. DCnVCs between SC and WC (**A**); DCnVCs between MC and SC (**B**); DCnVCs between MC and WC (**C**).

In the comparison between MC and SC, 103 DCnVCs were detected (Figure 2B), including organic acids and derivatives (27 nVCs, e.g., monoglyceride citrate, isochorismate, 4-hydroxystachydrine, etc.), lipids and lipid-like molecules (22 nVCs, e.g., di-

fluprednatum, pegorgotein, kojibiose, dehydroepiandrosterone, etc.), organic oxygen compounds (18 nVCs, e.g., etimicin, kelampayoside A, D-digitoxose, D-galactraric acid, etc.), organoheterocyclic compounds (15 nVCs, e.g., theophylline, castanospermine, ethofumesate, etc.), phenylpropanoids and polyketides (6 nVCs, e.g., isorhamnetin, vanilloyl glucose, zeranol, etc.), benzenoids (5 nVCs, e.g., 4-ethoxybenzamide, dopamine, methyl isoeugenol, etc.), nucleosides, nucleotides, and analogs (3 nVCs, 3-methyluridine, xanthosine, and adenosine-2′-phosphate), alkaloids and their derivatives (2 nVCs, pilocarpine and isoguvacine), organic nitrogen compounds (1 nVCs, N,N-dimethylsphingosine), and others (4 nVCs). The relative levels of 85 nVCs were decreased significantly, and those of 18 nVCs were increased significantly.

Conversely, only 20 DCnVCs were detected between MC and WC (Figure 2C). Among them, the relative levels of 8nVCs decreased significantly, while those of 12 nVCs increased significantly. These DCnVCs related to lipids and lipid-like molecules (7 nVCs, e.g., neoconvallatoxoloside, rosmaridiphenol, PA(18:1(9Z)/18:2(9Z,12Z0), physalin D, PA(18:1(9Z)/16:0), PI(16:1(9Z)/18:1(9Z)), etc.), organic acids and their derivatives (2 nVCs, imazamox and tyrosyl-phenylalanine), organic oxygen compounds (2 nVCs, CMP-2-aminoethylphosphonate and (S)-2-hfdroxy-2-phenylacetonitrile O-[b-D-apiosyl-(1-2)-b-D-glucoside]), alkaloids and their derivatives (1 nVCs, harmane), benzenoids (1 nVCs, tiapamil), organic nitrogen compounds (1 nVCs, agmatine), organoheterocyclic compounds (1 nVCs, talipexole), phenylpropanoids and polyketides (1 nVCs, 3-demethylsimmondsin 2′-(Z)-ferulate), and others (4 nVCs).

Based on the DCnVCs observed for different primary processing methods, MC/WC showed the least number of DCnVCs, while the DCnVCs between SC with WC and MC were higher in number compared to MC/WC. Moreover, the number of DCnVCs in SC/WC was the highest, followed by MS/SC, with the lowest number of DCnVCs observed in MC/WC. Notably, the nVCs glutamic acid and galactinol were found to be associated with the washed and honey processes [26]. Specifically, n-acetyl-L-glutamic acid and D-galactaric acid were identified as DCnVCs in SC/WC and SC/MC, while D-galactaric acid, glycine, lysine, sorbose, fructose, glyceric acid, and glycolic acid were associated with the dry process [26].

The impact of post-harvest coffee processing on the composition of coffee beans was found to be inconsistent in previous studies [2,27,28]. While the volatile constituents of green coffee beans showed no significant influence on coffee aroma composition, the associated metabolites, such as gluconic acid and sugar alcohols, tended to accumulate in the drying outer layers of the coffee cherries. Therefore, primary processing methods emerged as the dominant factor affecting coffee metabolite composition, significantly influencing the composition of the final green coffee beans, influenced in part by microbial community structures [2,26,27].

In the context of marker compounds for roasted beans resulting from natural, washed, and honey processing, six markers were identified, namely caffeoyl tryptophan, L-valine, guaiacol acetate, indol-3-acetyl-valine, coriandrone E, and picrocrocin. Additionally, seven markers, i.e., 2-C-methyl-D-erythrono-1,4-lactone, homoarecoline, naringenin, 2-methyl-1-phenyl-2-propanyl butyrate, 4-methoxysalicylic acid, 2-aminoheptanedioic acid, 2-methyl-3-(methylthio)furan, and five markers (N-feruloylglycine, 3-(2-furanylmethylene)pyrrolidine, 4-ethyl-2-methyloxazole, gluconic acid, and 2-acetyl-6-methylpyridine) were associated with post-harvest processing [28].

Furthermore, 1-O-caffeoylquinic acid was identified as an increased differential metabolite in the primary processing method between SC and WC. The phenolic composition of coffee is influenced by factors such as coffee species, cultivars, and the coffee roasting process [25]. Therefore, 1-O-caffeoylquinic acid can serve as a discriminant marker compound for distinguishing between washed and natural processing methods.

### 3.2. Analysis of Differentially Changed Volatile Compounds (DCVCs)

Volatile compounds play a crucial role in determining coffee quality [20]. A total of 176 volatile compounds (VCs) were detected in coffee samples from different processing methods, as shown in Table 1. These compounds were grouped into 19 classes, including acids (48 VCs), alcohols (28 VCs), hydrocarbons (22 VCs), ketones (20 VCs), esters (10 VCs), amides (8 VCs), aldehydes (5 VCs), pyridines (5 VCs), benzenoids (5 VCs), amines (5 VCs), lactone (4 VCs), amino acids (2 VCs), furanone (2 VCs), pyrimidines (2 VCs), piperidine (1 VCs), pyrazole (1 VCs), alkane (1 VCs), ether (1 VCs), and others (6 VCs). Among the volatile compounds present in WC samples, the top five were quininic acid, 3-[(tetrahydro-2H-pyran-2-yl)oxy]-benzenamine, 2-ethylhexanal ethylene glycol acetal, hexonic acid,3-deoxy-gamma-lactone, and sucrose. For SC samples, the top five compounds included quininic acid, 2-ethylhexanal ethylene glycol acetal, 3-[(tetrahydro-2H-pyran-2-yl)oxy]-benzenamine, sucrose, and D-(+)-trehalose. In MC samples, the major volatile compounds were quininic acid, 2-ethylhexanal ethylene glycol acetal, isochlorogenic acid, hexonic acid, 3-deoxy-gamma-lactone, and glycolic acid. Notably, quininic acid was the most abundant compound in all coffee samples, regardless of the primary processing method.

**Table 1.** Volatile compounds from different coffee primary processing methods.

| No. | Volatile Compounds | Class | RI | RT/min | Relative Level/% | | |
|-----|--------------------|-------|-----|--------|------|------|------|
| | | | | | WC | SC | MC |
| 1 | 4-hydroxy-Butanoic acid | acids | 922.76 | 6.23 | $0.03\% \pm 0.00\%$ | $0.01\% \pm 0.00\%$ | $0.01\% \pm 0.00\%$ |
| 2 | N,N,O-Triacetyl hydroxylamine | amines | 930.41 | 6.33 | $0.09\% \pm 0.01\%$ | $0.05\% \pm 0.01\%$ | $0.09\% \pm 0.02\%$ |
| 3 | N,N-diethyl-Formamide | amides | 945.39 | 6.54 | $0.05\% \pm 0.01\%$ | $0.04\% \pm 0.01\%$ | $0.03\% \pm 0.00\%$ |
| 4 | Dihydro-3-methylene-2,5-Furandione | furanones | 950.99 | 6.62 | $0.03\% \pm 0.00\%$ | $0.03\% \pm 0.01\%$ | $0.03\% \pm 0.01\%$ |
| 5 | Methanediimine | amines | 954.36 | 6.66 | $0.21\% \pm 0.07\%$ | $0.27\% \pm 0.09\%$ | $0.31\% \pm 0.02\%$ |
| 6 | Methoxyimino acetic acid | acids | 959.09 | 6.73 | $0.41\% \pm 0.02\%$ | $0.36\% \pm 0.05\%$ | $0.38\% \pm 0.02\%$ |
| 7 | 3-ethyl-Pyridine | pyridines | 974.13 | 6.95 | $0.01\% \pm 0.00\%$ | $0.01\% \pm 0.00\%$ | $0.01\% \pm 0.00\%$ |
| 8 | N-methyl-N-(2-methoxyethoxycarbonyl)-Alanine undecyl ester | esters | 987.08 | 7.15 | $0.04\% \pm 0.02\%$ | $0.05\% \pm 0.01\%$ | $0.05\% \pm 0.00\%$ |
| 9 | Propanoic acid,2-methyl-,3-phenylpropyl ester | esters | 994.39 | 7.26 | $0.04\% \pm 0.00\%$ | $0.03\% \pm 0.00\%$ | $0.03\% \pm 0.00\%$ |
| 10 | Furfuryl alcohol | alcohols | 996.44 | 7.29 | $1.82\% \pm 0.17\%$ | $0.80\% \pm 0.17\%$ | $1.54\% \pm 0.27\%$ |
| 11 | 3,3-Dimethylacrylic acid | acids | 1009.54 | 7.50 | $0.02\% \pm 0.00\%$ | $0.01\% \pm 0.00\%$ | $0.02\% \pm 0.00\%$ |
| 12 | 2-Aminoethanol | alcohols | 1023.06 | 7.72 | $0.03\% \pm 0.05\%$ | $0.01\% \pm 0.01\%$ | $0.00\% \pm 0.00\%$ |
| 13 | 2,3-Butanediol | alcohols | 1037.43 | 7.97 | $0.04\% \pm 0.00\%$ | $0.36\% \pm 0.02\%$ | $0.10\% \pm 0.01\%$ |
| 14 | 2-Hydroxypyridine | pyridines | 1038.13 | 7.98 | $0.84\% \pm 0.66\%$ | $1.31\% \pm 0.38\%$ | $1.51\% \pm 0.11\%$ |
| 15 | Methyl leucinate | esters | 1038.78 | 7.99 | $0.02\% \pm 0.00\%$ | $0.01\% \pm 0.00\%$ | $0.02\% \pm 0.00\%$ |
| 16 | Pyruvic acid | acids | 1048.13 | 8.15 | $0.23\% \pm 0.05\%$ | $0.43\% \pm 0.01\%$ | $0.26\% \pm 0.05\%$ |
| 17 | Lactic acid | acids | 1057.58 | 8.32 | $3.57\% \pm 0.19\%$ | $3.13\% \pm 0.07\%$ | $3.36\% \pm 0.07\%$ |
| 18 | Glycolic acid | acids | 1074.89 | 8.63 | $3.83\% \pm 0.30\%$ | $3.56\% \pm 0.06\%$ | $3.86\% \pm 0.06\%$ |
| 19 | 2-Pyrrolidinone | ketones | 1080.90 | 8.74 | $0.01\% \pm 0.00\%$ | $0.04\% \pm 0.01\%$ | $0.01\% \pm 0.00\%$ |
| 20 | 1-(5-Dimethylethyl)pyrazin-2-yl-ethan-1-one | ketones | 1085.67 | 8.83 | $0.00\% \pm 0.00\%$ | $0.00\% \pm 0.00\%$ | $0.00\% \pm 0.00\%$ |
| 21 | 4-Oxo-4,5,6,7-tetrahydrobenzofuroxan | others | 1095.32 | 9.02 | $0.03\% \pm 0.01\%$ | $0.02\% \pm 0.01\%$ | $0.03\% \pm 0.01\%$ |
| 22 | 2-Hydroxybutyric acid | acids | 1122.19 | 9.48 | $0.15\% \pm 0.01\%$ | $0.11\% \pm 0.01\%$ | $0.16\% \pm 0.01\%$ |
| 23 | 2-Furoic acid | acids | 1136.53 | 9.73 | $0.40\% \pm 0.05\%$ | $0.34\% \pm 0.03\%$ | $0.44\% \pm 0.03\%$ |
| 24 | 3-Pyridinol | alcohols | 1141.15 | 9.81 | $3.69\% \pm 0.18\%$ | $2.93\% \pm 0.16\%$ | $3.65\% \pm 0.09\%$ |
| 25 | 3-Hydroxypropionic acid | acids | 1141.72 | 9.82 | $0.64\% \pm 0.01\%$ | $0.78\% \pm 0.02\%$ | $0.66\% \pm 0.04\%$ |
| 26 | Methyl nicotinate | esters | 1150.51 | 9.97 | $0.02\% \pm 0.00\%$ | $0.01\% \pm 0.00\%$ | $0.02\% \pm 0.01\%$ |
| 27 | 1-Piperidinecarboxaldehyde | aldehydes | 1154.52 | 10.05 | $0.03\% \pm 0.00\%$ | $0.01\% \pm 0.01\%$ | $0.00\% \pm 0.00\%$ |
| 28 | Acetaldehyde, tetramer | aldehydes | 1163.98 | 10.22 | $0.01\% \pm 0.00\%$ | $0.00\% \pm 0.00\%$ | $0.01\% \pm 0.00\%$ |
| 29 | Butan-1-ol | alcohols | 1167.68 | 10.29 | $0.19\% \pm 0.01\%$ | $0.15\% \pm 0.01\%$ | $0.16\% \pm 0.00\%$ |
| 30 | 3-(1-Hydroxy-1-methylethyl)benzonitrile | benzenoids | 1173.66 | 10.40 | $0.05\% \pm 0.01\%$ | $0.02\% \pm 0.04\%$ | $0.00\% \pm 0.00\%$ |

**Table 1.** *Cont.*

| No. | Volatile Compounds | Class | RI | RT/min | Relative Level/% | | |
|---|---|---|---|---|---|---|---|
| | | | | | WC | SC | MC |
| 31 | 1-methoxy-4-phenoxy-Benzene | benzenoids | 1183.02 | 10.57 | 0.01% ± 0.00% | 0.01% ± 0.00% | 0.01% ± 0.00% |
| 32 | 1,3-benzodioxole-5-carboxylic acid | acids | 1185.07 | 10.61 | 0.01% ± 0.00% | 0.02% ± 0.00% | 0.02% ± 0.00% |
| 33 | 1H-pyrimidine-2,4-dione | ketones | 1200.12 | 10.90 | 0.05% ± 0.00% | 0.03% ± 0.01% | 0.03% ± 0.01% |
| 34 | 5-Hydroxy-2-methylpyridine | pyridines | 1201.35 | 10.92 | 0.25% ± 0.01% | 0.19% ± 0.01% | 0.24% ± 0.01% |
| 35 | Phloroglucinol | alcohols | 1207.11 | 11.02 | 0.07% ± 0.00% | 0.08% ± 0.01% | 0.05% ± 0.00% |
| 36 | 1-t-Butyldioxymethyl-4-methylpiperidine | piperidines | 1210.09 | 11.07 | 0.03% ± 0.00% | 0.01% ± 0.01% | 0.00% ± 0.00% |
| 37 | M-Aminophenylacetylene | others | 1217.76 | 11.19 | 0.26% ± 0.02% | 0.32% ± 0.01% | 0.25% ± 0.02% |
| 38 | Guaicol | alcohols | 1227.35 | 11.35 | 0.02% ± 0.01% | 0.01% ± 0.00% | 0.01% ± 0.00% |
| 39 | 4-Hydroxybutanoic acid | acids | 1232.42 | 11.44 | 0.08% ± 0.01% | 0.08% ± 0.01% | 0.09% ± 0.00% |
| 40 | 2-Ketoadipic acid | acids | 1242.56 | 11.61 | 0.06% ± 0.01% | 0.04% ± 0.00% | 0.07% ± 0.01% |
| 41 | N-ethyl-3,5-di(hydroxymethyl)-Aniline | amines | 1243.48 | 11.63 | 0.05% ± 0.01% | 0.04% ± 0.00% | 0.04% ± 0.00% |
| 42 | Benzoic acid | acids | 1250.20 | 11.75 | 0.01% ± 0.00% | 0.01% ± 0.00% | 0.01% ± 0.00% |
| 43 | N-Acetyl Alanine | amino acids | 1261.54 | 11.95 | 0.02% ± 0.00% | 0.02% ± 0.00% | 0.02% ± 0.00% |
| 44 | Glycerol | alcohols | 1266.42 | 12.03 | 0.36% ± 0.05% | 1.31% ± 0.12% | 0.40% ± 0.01% |
| 45 | 4-(Hydrazinylmethyl)-1-methylpyrazole | pyrazoles | 1270.59 | 12.11 | 0.02% ± 0.00% | 0.01% ± 0.00% | 0.02% ± 0.02% |
| 46 | Maltol | alcohols | 1288.05 | 12.43 | 0.63% ± 0.06% | 0.14% ± 0.18% | 0.05% ± 0.01% |
| 47 | 2-fluoro-3-hydroxy-4-methoxy-Benzaldehyde | aldehydes | 1288.87 | 12.44 | 0.03% ± 0.00% | 0.03% ± 0.00% | 0.03% ± 0.00% |
| 48 | 2-Oxovaleric acid | acids | 1294.19 | 12.54 | 0.16% ± 0.01% | 0.14% ± 0.01% | 0.14% ± 0.00% |
| 49 | Nicotinic acid | acids | 1295.26 | 12.56 | 2.18% ± 0.00% | 2.32% ± 0.10% | 2.21% ± 0.11% |
| 50 | Succinic acid | acids | 1308.61 | 12.79 | 0.29% ± 0.11% | 0.34% ± 0.04% | 0.36% ± 0.02% |
| 51 | Catechol | alcohols | 1313.67 | 12.87 | 0.64% ± 0.08% | 0.42% ± 0.04% | 0.61% ± 0.04% |
| 52 | Glyceric acid | acids | 1322.07 | 13.00 | 0.52% ± 0.13% | 0.62% ± 0.06% | 0.71% ± 0.02% |
| 53 | Fumaric acid | acids | 1331.01 | 13.14 | 0.07% ± 0.01% | 0.08% ± 0.02% | 0.06% ± 0.01% |
| 54 | Itaconic acid | acids | 1337.71 | 13.25 | 0.73% ± 0.06% | 0.62% ± 0.04% | 0.75% ± 0.02% |
| 55 | Citraconic acid | acids | 1345.20 | 13.38 | 0.93% ± 0.33% | 0.84% ± 0.21% | 1.17% ± 0.09% |
| 56 | 2′-Hydroxy-5′-methylacetophenone | ketones | 1356.64 | 13.57 | 0.01% ± 0.00% | 0.01% ± 0.00% | 0.01% ± 0.00% |
| 57 | 1-(2-hydroxyphenyl)-2-phenylethanone | ketones | 1359.98 | 13.62 | 0.04% ± 0.00% | 0.04% ± 0.00% | 0.04% ± 0.00% |
| 58 | 2-Cyano-5-(4-fluorophenyl)pyrimidine | pyrimidines | 1374.09 | 13.86 | 0.01% ± 0.00% | 0.03% ± 0.01% | 0.01% ± 0.00% |
| 59 | 4-Methylcatechol | alcohols | 1385.71 | 14.06 | 0.07% ± 0.06% | 0.32% ± 0.24% | 0.07% ± 0.01% |
| 60 | 5-Hydroxymethylfurfural | furanone | 1398.92 | 14.29 | 0.61% ± 0.02% | 0.98% ± 0.07% | 0.63% ± 0.08% |
| 61 | Pentane-1,2,5-triol | alcohols | 1407.64 | 14.43 | 0.20% ± 0.02% | 0.07% ± 0.01% | 0.01% ± 0.01% |
| 62 | (Z)-Erythrono-1,4-lactone | lactones | 1424.24 | 14.68 | 0.04% ± 0.01% | 0.08% ± 0.00% | 0.05% ± 0.01% |
| 63 | 4′-Hydroxy-3′-methoxyacetophenone | ketones | 1445.73 | 15.01 | 0.01% ± 0.00% | 0.01% ± 0.00% | 0.01% ± 0.00% |
| 64 | 6-Cyano-5-methyl-1,3-diazaadamantan-6-ol | alcohols | 1454.84 | 15.16 | 0.01% ± 0.00% | 0.01% ± 0.00% | 0.41% ± 0.02% |
| 65 | 3-Methylorsellinic acid | acids | 1456.54 | 15.18 | 0.13% ± 0.02% | 0.09% ± 0.02% | 0.12% ± 0.01% |
| 66 | Methylhydroquinone | ketones | 1457.93 | 15.21 | 0.03% ± 0.00% | 0.02% ± 0.00% | 0.03% ± 0.00% |
| 67 | 2,2′-Bipyridine | pyridines | 1459.31 | 15.23 | 0.03% ± 0.00% | 0.04% ± 0.01% | 0.04% ± 0.00% |
| 68 | Pipecolic acid | acids | 1470.15 | 15.40 | 0.33% ± 0.01% | 0.22% ± 0.02% | 0.23% ± 0.01% |
| 69 | Malic acid | acids | 1480.39 | 15.57 | 1.99% ± 0.73% | 2.67% ± 0.42% | 2.84% ± 0.11% |
| 70 | Tropic Acid | acids | 1503.23 | 15.94 | 0.86% ± 0.36% | 0.89% ± 0.17% | 0.74% ± 0.39% |
| 71 | Pyroglutamic acid | acids | 1509.09 | 16.02 | 2.92% ± 0.23% | 1.94% ± 0.06% | 2.49% ± 0.10% |
| 72 | 4-pentyl-1,1′-Biphenyl | benzenoids | 1512.50 | 16.07 | 0.04% ± 0.00% | 0.04% ± 0.00% | 0.04% ± 0.00% |
| 73 | Acetoisovanillone | ketones | 1520.04 | 16.18 | 0.01% ± 0.00% | 0.01% ± 0.00% | 0.01% ± 0.00% |
| 74 | 2-Aminobenzophenone | ketones | 1535.38 | 16.40 | 0.30% ± 0.01% | 0.43% ± 0.02% | 0.29% ± 0.02% |
| 75 | Pyrogallol | alcohols | 1536.20 | 16.41 | 0.49% ± 0.03% | 0.39% ± 0.03% | 0.43% ± 0.01% |
| 76 | Methylsuccinic acid | acids | 1542.43 | 16.51 | 0.11% ± 0.13% | 0.14% ± 0.17% | 0.03% ± 0.00% |
| 77 | 5-Hydroxymethyl-2-furoic acid | acids | 1546.75 | 16.57 | 0.25% ± 0.04% | 0.33% ± 0.02% | 0.31% ± 0.01% |
| 78 | 3-Hydroxybenzoate | benzenoids | 1563.86 | 16.83 | 0.16% ± 0.02% | 0.16% ± 0.01% | 0.16% ± 0.01% |
| 79 | 4-(2-Hydroxy-5-Nitrophenyl)Pyrimidine | pyrimidines | 1587.80 | 17.19 | 0.03% ± 0.00% | 0.02% ± 0.00% | 0.03% ± 0.00% |
| 80 | 1,2,4-Benzenetriol | alcohols | 1594.59 | 17.30 | 0.66% ± 0.10% | 0.51% ± 0.06% | 0.60% ± 0.03% |
| 81 | Arabinofuranose | hydrocarbons | 1599.55 | 17.37 | 0.20% ± 0.04% | 0.20% ± 0.01% | 0.19% ± 0.03% |
| 82 | D-(−)-Ribofuranose | hydrocarbons | 1608.25 | 17.49 | 0.10% ± 0.01% | 0.16% ± 0.02% | 0.11% ± 0.02% |
| 83 | 6-Hydroxynicotinic acid | acids | 1621.94 | 17.68 | 0.09% ± 0.00% | 0.10% ± 0.02% | 0.07% ± 0.01% |

<div align="center"><b>Table 1.</b> <i>Cont.</i></div>

| No. | Volatile Compounds | Class | RI | RT/min | Relative Level/% | | |
|---|---|---|---|---|---|---|---|
| | | | | | WC | SC | MC |
| 84 | 3,4-dihydroxy-5-(hydroxymethyl)-3-methyloxolan-2-one | ketones | 1632.19 | 17.83 | 0.03% ± 0.02% | 0.06% ± 0.02% | 0.03% ± 0.01% |
| 85 | Ribose | hydrocarbons | 1643.92 | 17.99 | 0.03% ± 0.01% | 0.29% ± 0.42% | 0.04% ± 0.00% |
| 86 | Vanillin | aldehydes | 1646.19 | 18.02 | 0.01% ± 0.00% | 0.02% ± 0.00% | 0.02% ± 0.00% |
| 87 | D-Lyxose | hydrocarbons | 1651.05 | 18.09 | 0.44% ± 0.02% | 0.73% ± 0.12% | 0.49% ± 0.01% |
| 88 | Methylalpha-Lyxofuranoside | esters | 1654.92 | 18.15 | 0.17% ± 0.02% | 0.10% ± 0.02% | 0.17% ± 0.01% |
| 89 | D-Xylulose | hydrocarbons | 1664.70 | 18.29 | 0.49% ± 0.02% | 0.70% ± 0.07% | 0.54% ± 0.05% |
| 90 | Xylose | hydrocarbons | 1665.54 | 18.30 | 0.17% ± 0.01% | 0.27% ± 0.02% | 0.18% ± 0.02% |
| 91 | D-(+)-Ribono-1,4-lactone | lactones | 1674.41 | 18.43 | 0.29% ± 0.10% | 0.25% ± 0.07% | 0.24% ± 0.05% |
| 92 | 1,6-Anhydro-Glucose | hydrocarbons | 1695.11 | 18.73 | 0.23% ± 0.01% | 0.17% ± 0.01% | 0.21% ± 0.00% |
| 93 | Fucose | hydrocarbons | 1704.90 | 18.86 | 0.06% ± 0.00% | 0.07% ± 0.01% | 0.06% ± 0.01% |
| 94 | Gallacetophenone-4′-methylether | others | 1705.66 | 18.87 | 0.13% ± 0.04% | 0.15% ± 0.02% | 0.15% ± 0.01% |
| 95 | Udp-Glucuronic acid | acids | 1732.16 | 19.22 | 0.11% ± 0.00% | 0.10% ± 0.00% | 0.11% ± 0.00% |
| 96 | Glycerol 1-Phosphate | esters | 1755.15 | 19.53 | 0.12% ± 0.05% | 0.10% ± 0.01% | 0.12% ± 0.01% |
| 97 | 5-(1,2-dihydroxyethyl)-3-hydroxyoxolan-2-one | ketones | 1769.99 | 19.74 | 1.58% ± 0.13% | 1.35% ± 0.10% | 1.59% ± 0.03% |
| 98 | Hexonic acid,3-deoxy-gamma-lactone | lactones | 1775.68 | 19.81 | 4.67% ± 0.50% | 3.32% ± 0.43% | 4.18% ± 0.61% |
| 99 | L-Iditol | alcohols | 1787.21 | 19.97 | 1.90% ± 0.15% | 1.48% ± 0.10% | 2.00% ± 0.06% |
| 100 | Shikimic acid | acids | 1798.42 | 20.13 | 0.15% ± 0.02% | 0.13% ± 0.01% | 0.14% ± 0.01% |
| 101 | 1,2,4,5-Cyclohexanetetrol | alcohols | 1804.75 | 20.21 | 0.17% ± 0.02% | 0.14% ± 0.02% | 0.20% ± 0.01% |
| 102 | Citric acid | acids | 1809.00 | 20.26 | 0.23% ± 0.17% | 0.50% ± 0.30% | 0.73% ± 0.18% |
| 103 | 3′-Methyl-2-benzylidene-coumaran-3-one | ketones | 1813.56 | 20.32 | 0.10% ± 0.01% | 0.09% ± 0.00% | 0.08% ± 0.01% |
| 104 | Protocatechuic acid | acids | 1814.60 | 20.33 | 0.10% ± 0.01% | 0.11% ± 0.01% | 0.11% ± 0.01% |
| 105 | Quinic acid | acids | 1827.25 | 20.49 | 0.18% ± 0.02% | 0.12% ± 0.05% | 0.20% ± 0.02% |
| 106 | Cyclo(L-prolyl-L-valine) | others | 1832.07 | 20.55 | 0.03% ± 0.00% | 0.04% ± 0.00% | 0.03% ± 0.00% |
| 107 | Quininic acid | acids | 1850.07 | 20.78 | 6.15% ± 1.13% | 6.94% ± 0.18% | 7.56% ± 0.31% |
| 108 | 2-Ethylhexanal ethylene glycol acetal | aldehydes | 1853.42 | 20.82 | 5.53% ± 0.67% | 5.53% ± 0.18% | 6.06% ± 0.33% |
| 109 | Methyl-Urea | amino acids | 1853.73 | 20.83 | 0.52% ± 0.05% | 0.43% ± 0.18% | 0.57% ± 0.02% |
| 110 | Fructose | hydrocarbons | 1857.71 | 20.88 | 0.14% ± 0.06% | 0.19% ± 0.04% | 0.05% ± 0.01% |
| 111 | Tagatose | hydrocarbons | 1861.12 | 20.92 | 0.09% ± 0.08% | 0.45% ± 0.12% | 0.10% ± 0.03% |
| 112 | Sorbose | hydrocarbons | 1869.17 | 21.03 | 0.11% ± 0.02% | 0.38% ± 0.09% | 0.11% ± 0.02% |
| 113 | Galactose | hydrocarbons | 1874.19 | 21.09 | 0.09% ± 0.00% | 0.28% ± 0.07% | 0.10% ± 0.01% |
| 114 | 3-[(tetrahydro-2H-pyran-2-yl)oxy]-Benzenamine | amines | 1877.11 | 21.13 | 6.06% ± 3.43% | 5.47% ± 2.27% | 2.29% ± 0.24% |
| 115 | D-Allose | hydrocarbons | 1878.88 | 21.15 | 0.24% ± 0.01% | 0.31% ± 0.02% | 0.24% ± 0.02% |
| 116 | D-(+)-Altrose | hydrocarbons | 1884.93 | 21.23 | 0.21% ± 0.03% | 0.78% ± 0.22% | 0.17% ± 0.04% |
| 117 | 4,N-dipropyl-Benzamide | amides | 1901.50 | 21.45 | 0.05% ± 0.00% | 0.06% ± 0.00% | 0.06% ± 0.02% |
| 118 | D-Mannitol | alcohols | 1918.22 | 21.65 | 2.17% ± 0.04% | 2.12% ± 0.21% | 2.38% ± 0.59% |
| 119 | Ethylalpha-D-glucopyranoside | hydrocarbons | 1922.52 | 21.70 | 0.10% ± 0.03% | 0.76% ± 0.02% | 0.07% ± 0.01% |
| 120 | N-propargyloxycarbonyl-L-Norvaline pentyl ester | esters | 1928.78 | 21.78 | 0.02% ± 0.02% | 0.02% ± 0.00% | 0.02% ± 0.00% |
| 121 | Hexahydro-3-(2-methylpropyl)-Pyrrolo[1,2-a]pyrazine-1,4-dione | ketones | 1955.16 | 22.10 | 0.14% ± 0.01% | 0.11% ± 0.01% | 0.11% ± 0.00% |
| 122 | D-Glucose | hydrocarbons | 1962.12 | 22.19 | 0.08% ± 0.01% | 0.09% ± 0.02% | 0.09% ± 0.01% |
| 123 | Mannonic acid, gamma-lactone | lactones | 1968.97 | 22.27 | 0.08% ± 0.01% | 0.22% ± 0.03% | 0.20% ± 0.02% |
| 124 | 5-Hydroxy-7-methoxy-2-methyl-3-phenyl-4-chromenone | ketones | 1974.52 | 22.34 | 0.10% ± 0.01% | 0.13% ± 0.01% | 0.09% ± 0.02% |
| 125 | Hexitol | alcohols | 1982.52 | 22.44 | 0.04% ± 0.00% | 0.02% ± 0.00% | 0.04% ± 0.01% |
| 126 | 17-Methoxy-d-homo-18-norandrosta-4,8,13,15,17-pentaen-3-one | ketones | 2016.32 | 22.85 | 0.01% ± 0.00% | 0.01% ± 0.00% | 0.01% ± 0.00% |
| 127 | Palmitic acid | acids | 2043.35 | 23.16 | 1.36% ± 0.22% | 1.26% ± 0.09% | 1.54% ± 0.08% |
| 128 | Myo-Inositol | alcohols | 2080.99 | 23.60 | 3.22% ± 0.28% | 3.22% ± 0.05% | 3.23% ± 0.09% |
| 129 | Caffeic acid | acids | 2131.93 | 24.18 | 0.55% ± 0.35% | 0.81% ± 0.09% | 0.80% ± 0.10% |
| 130 | Hexadecanamide | amides | 2183.03 | 24.75 | 0.23% ± 0.03% | 0.20% ± 0.03% | 0.22% ± 0.02% |

Table 1. *Cont.*

| No. | Volatile Compounds | Class | RI | RT/min | Relative Level/% | | |
|---|---|---|---|---|---|---|---|
| | | | | | WC | SC | MC |
| 131 | 5-Propoxy-2,2′-bipyridyl | pyridines | 2199.56 | 24.94 | 0.01% ± 0.00% | 0.01% ± 0.00% | 0.01% ± 0.00% |
| 132 | Linoleic acid | acids | 2209.15 | 25.04 | 0.26% ± 0.10% | 0.23% ± 0.10% | 0.36% ± 0.03% |
| 133 | (Z)-Oleic acid | acids | 2214.84 | 25.10 | 0.18% ± 0.02% | 0.14% ± 0.04% | 0.14% ± 0.04% |
| 134 | Stearic acid | acids | 2240.22 | 25.37 | 0.25% ± 0.02% | 0.24% ± 0.01% | 0.26% ± 0.01% |
| 135 | 2,3,6,7,8,8a-hexahydro-1,4-dioxo-Pyrrolo[1,2-a]pyrazine-3-propanamide, | amides | 2272.20 | 25.71 | 0.16% ± 0.00% | 0.13% ± 0.01% | 0.12% ± 0.02% |
| 136 | Hexahydro-3-(phenylmethyl)-Pyrrolo[1,2-a]pyrazine-1,4-dione | ketones | 2360.27 | 26.63 | 0.04% ± 0.01% | 0.03% ± 0.01% | 0.03% ± 0.00% |
| 137 | (E,E)-9,12-Octadecadienoic acid, methyl ester | esters | 2362.49 | 26.65 | 0.05% ± 0.00% | 0.04% ± 0.00% | 0.04% ± 0.00% |
| 138 | (Z)-9-Octadecenamide | amides | 2368.57 | 26.71 | 0.41% ± 0.01% | 0.33% ± 0.12% | 0.33% ± 0.07% |
| 139 | (Z)-13-Docosenamide | amides | 2373.58 | 26.76 | 0.19% ± 0.05% | 0.31% ± 0.28% | 0.15% ± 0.02% |
| 140 | 9-(2-p-Tolylethyl)-3,4,5,6,7,9-hexahydro-2H-xanthene-1,8-dione | ketones | 2384.53 | 26.88 | 0.24% ± 0.07% | 0.19% ± 0.02% | 0.25% ± 0.02% |
| 141 | Octadecanamide | amides | 2391.42 | 26.95 | 0.11% ± 0.02% | 0.09% ± 0.03% | 0.09% ± 0.02% |
| 142 | D-Erythro-Sphingosine | alcohols | 2413.58 | 27.17 | 0.08% ± 0.01% | 0.05% ± 0.01% | 0.07% ± 0.01% |
| 143 | Arachidic acid | acids | 2437.35 | 27.40 | 0.09% ± 0.01% | 0.09% ± 0.00% | 0.10% ± 0.00% |
| 144 | Uridine | others | 2444.21 | 27.47 | 0.23% ± 0.04% | 0.18% ± 0.01% | 0.23% ± 0.03% |
| 145 | 1beta,12,12-trimethyl-7,11-Dioxapentacyclo[15.3.0.0(4,16).0(5,13).0(5,10)]eicos-13-en-20-ol-8-one | ketones | 2463.98 | 27.66 | 0.07% ± 0.00% | 0.06% ± 0.00% | 0.06% ± 0.00% |
| 146 | 8-Benzylquinoline | others | 2521.86 | 28.23 | 0.04% ± 0.00% | 0.04% ± 0.00% | 0.04% ± 0.00% |
| 147 | Estrone | ketones | 2546.41 | 28.46 | 0.06% ± 0.01% | 0.05% ± 0.02% | 0.05% ± 0.01% |
| 148 | Arbutin | others | 2565.43 | 28.64 | 0.03% ± 0.01% | 0.03% ± 0.00% | 0.03% ± 0.00% |
| 149 | Nonanamide | amides | 2600.22 | 28.97 | 0.02% ± 0.00% | 0.02% ± 0.00% | 0.02% ± 0.00% |
| 150 | D-(+)-Trehalose | hydrocarbons | 2604.85 | 29.01 | 4.20% ± 0.06% | 4.87% ± 0.22% | 3.80% ± 0.44% |
| 151 | 2,2-Bis(3-allyl-4-hydroxyphenyl)propane | alkanes | 2620.91 | 29.16 | 0.76% ± 0.04% | 0.90% ± 0.03% | 0.69% ± 0.08% |
| 152 | Sucrose | hydrocarbons | 2621.57 | 29.17 | 4.27% ± 0.08% | 4.96% ± 0.28% | 3.84% ± 0.46% |
| 153 | Terephthalic acid,cyclobutyl decyl ester | esters | 2626.50 | 29.21 | 0.11% ± 0.08% | 0.12% ± 0.08% | 0.05% ± 0.03% |
| 154 | Trehalose | hydrocarbons | 2672.92 | 29.64 | 0.05% ± 0.00% | 0.05% ± 0.01% | 0.04% ± 0.00% |
| 155 | 1-(5-ethyl-2-hydroxy-4-methoxyphenyl)-2-(3,4-methylenedioxyphenyl)-Ethanone | ketones | 2688.21 | 29.78 | 0.05% ± 0.00% | 0.05% ± 0.00% | 0.04% ± 0.00% |
| 156 | 2-linoleoylglycerol | alcohols | 2709.56 | 29.97 | 0.03% ± 0.00% | 0.03% ± 0.00% | 0.03% ± 0.00% |
| 157 | Gentiobiose | hydrocarbons | 2728.82 | 30.14 | 0.02% ± 0.00% | 0.04% ± 0.00% | 0.02% ± 0.00% |
| 158 | 2-Oleoylglycerol | alcohols | 2746.27 | 30.29 | 0.12% ± 0.05% | 0.09% ± 0.02% | 0.08% ± 0.01% |
| 159 | Beta-D-Lactose | hydrocarbons | 2787.32 | 30.66 | 1.52% ± 0.26% | 1.03% ± 0.21% | 1.69% ± 0.25% |
| 160 | Beta-Gentiobiose | hydrocarbons | 2828.99 | 31.01 | 0.04% ± 0.01% | 0.04% ± 0.01% | 0.03% ± 0.01% |
| 161 | Galactinol | alcohols | 2970.75 | 32.21 | 3.60% ± 0.14% | 2.43% ± 0.14% | 3.47% ± 0.25% |
| 162 | 3-Methylbenzoic acid,2,5-dichlorophenyl ester | esters | 2972.52 | 32.23 | 0.05% ± 0.00% | 0.05% ± 0.00% | 0.06% ± 0.00% |
| 163 | Beta-Tocopherol | alcohols | 2984.46 | 32.33 | 0.36% ± 0.01% | 0.32% ± 0.03% | 0.30% ± 0.01% |
| 164 | 4-O-Coumaroyl-D-quinic acid | acids | 3034.08 | 32.77 | 0.07% ± 0.01% | 0.09% ± 0.01% | 0.08% ± 0.01% |
| 165 | 3-O-Coumaroyl-D-quinic acid | acids | 3057.44 | 32.99 | 0.07% ± 0.03% | 0.08% ± 0.01% | 0.07% ± 0.00% |
| 166 | Cis-5-O-Feruloylquinic acid | acids | 3085.02 | 33.25 | 2.37% ± 0.20% | 2.50% ± 0.05% | 2.65% ± 0.04% |
| 167 | Pentamethylbenzene | benzenoids | 3110.04 | 33.49 | 0.32% ± 0.02% | 0.34% ± 0.01% | 0.33% ± 0.01% |
| 168 | 2-PhenY1Pyrrolo(2,1-B)benzothiazol | alcohols | 3128.51 | 33.69 | 0.73% ± 0.02% | 0.63% ± 0.05% | 0.73% ± 0.02% |
| 169 | A-Tocopherol | alcohols | 3140.47 | 33.81 | 0.04% ± 0.00% | 0.04% ± 0.00% | 0.03% ± 0.00% |
| 170 | 4-O-Feruloylquinic acid | acids | 3151.82 | 33.93 | 0.82% ± 0.22% | 1.04% ± 0.03% | 1.03% ± 0.01% |
| 171 | 3-O-Feruloylquinic acid | acids | 3180.34 | 34.23 | 0.69% ± 0.12% | 0.96% ± 0.10% | 0.86% ± 0.07% |
| 172 | Isochlorogenic acid | acids | 3194.27 | 34.38 | 4.13% ± 1.56% | 3.87% ± 1.92% | 5.51% ± 0.11% |
| 173 | Trans Caftaric acid | acids | 3223.01 | 34.72 | 0.05% ± 0.03% | 0.01% ± 0.00% | 0.02% ± 0.01% |
| 174 | Campesterol | alcohols | 3264.99 | 35.23 | 0.09% ± 0.01% | 0.07% ± 0.01% | 0.08% ± 0.00% |
| 175 | Phenanthro[9,10-b]quinoxaline-11-carboxylic acid | acids | 3268.41 | 35.27 | 0.48% ± 0.04% | 0.50% ± 0.07% | 0.49% ± 0.01% |
| 176 | Benzalaniline | amines | 1737.05 | 35.76 | 0.01% ± 0.00% | 0.01% ± 0.00% | 0.01% ± 0.00% |

Roasted coffee beans are rich in volatile compounds, including hydrocarbons, alcohols, aldehydes, ketones, carboxylic acids, esters, pyrazines, pyridines, sulfur compounds, furans, furanones, and phenols, among others. However, only a small number of volatile compounds significantly contribute to coffee flavor and aroma characteristics, such as furans, furanones, phenolic compounds, sulfur-containing compounds, and pyrazines [20,29]. Acids, which are the most abundant class of compounds, are influenced by the processing method and other factors [30], which significantly affect the acidity and complex taste [31]. Special acids can be formed under different post-treatments [31], with quinic acid and formic acid showing significant differences in anaerobic natural, honey, washed, and hot-airing dry processes [31]. Additionally, quinic acid was positively correlated with the coffee's body score [32], while furanone contributes to sweetness, brownness, bread, and caramel notes [29], while vanillin imparts a vanilla-like character [29].

The volatile characteristics of coffee are closely related to cultivars, processing styles, geographical origins, and processing techniques [19]. Figure 3 depicts an overview of DCVCs between SC, WC, and MC. In comparison of SC to WC, the relative levels of 15 VCs decreased significantly (Figure 3A), including five alcohols (maltol, catechol, furfuryl alcohol, and pentane-1,2,5-triol, D-erythro-sphingosine), two acids (pyroglutamic acid and 3,3-dimethylacrylic acid), one amino acid (methyl leucinate), one pyrimidine (4-(2-hydroxy-5-nitrophenyl)pyrimidine), one peridine ((1-t-butyldioxymethyl-4-methylpiperidine), one ketone (1H-pyrimidine-2,4-dione), one ester (methylalpha-lyxofuranoside), one pyridine (3-ethyl-pyridine), one amine (N,N,O-triacetylhydroxylamine), and one aldehyde (acetaldehyde, tetramer). Conversely, the relative levels of 17 VCs in SC/WC increased significantly, which included nine hydrocarbons (xylose, D-(−)-ribofuranose, D-lyxose, galactose, sorbose, D-(+)-altrose, tagatose, gentiobiose, ethylalpha-D-glucopyranoside), two lactones ((Z)-erythrono-1,4-lactone, mannonic acid,gamma-lactone), two alcohols (glycerol, 2-2,3-butanediol), one acid (pyruvic acid), one ketone (pyrrolidinone), one furanone (5-hydroxymethylfurfural), and one pyrimidine (2-cyano-5-(4-fluorophenyl)pyrimidine).

Between MS and SC, 25 DCVCs were detected (Figure 3B). Among these compounds, 16 significantly decreased volatile compounds, including nine hydrocarbons (xylose, D-(+)-altrose, D-lyxose, gentiobiose, galactose, sorbose, fructose, tagatose, ethylalpha-D-glucopyranoside), three alcohols (glycerol, 2,3-butanediol, phloroglucinol), one ketone (2-pyrrolidinone), one furanone (5-hydroxymethylfurfural), one pyrimidine (2-cyano-5-(4-fluorophenyl)pyrimidine), and one acid (pyruvic acid). Nine volatile compounds significantly increased, including one hydrocarbon *(beta*-D-lactose), one acid (3,3-dimethylacrylic acid), two alcohols (furfuryl alcohol, pentane-1,2,5-triol), one aldehyde (acetaldehyde, tetramer), one pyridine (3-methyl-pyridine), one ester (methylalpha-lyxofuranoside), one amine (N,N,O-triacetylhydroxylamine), and one pyrimidine (4-(2-hydroxy-5-nitrophenyl)pyrimidine).

In the comparison between MC and WC, only nine DCVCs were detected (Figure 3C). Among these, six volatile compounds, including one alcohol (maltol), one ketone (1H-pyrimidine-2,4-dione), one amide (N,N-diethyl-formamide), one aldehyde (1-piperidine carboxaldehyde), one piperidine (1-t-butyldioxymethyl-4-methylpiperidine), and one benzenoid (3-(1-hydroxy-1-methylethyl)benzonitrile), decreased significantly. Three volatile compounds, including one alcohol (2,3-butanediol), one acid (citric acid), and one lactone (mannonic acid, gamma-lactone), increased significantly.

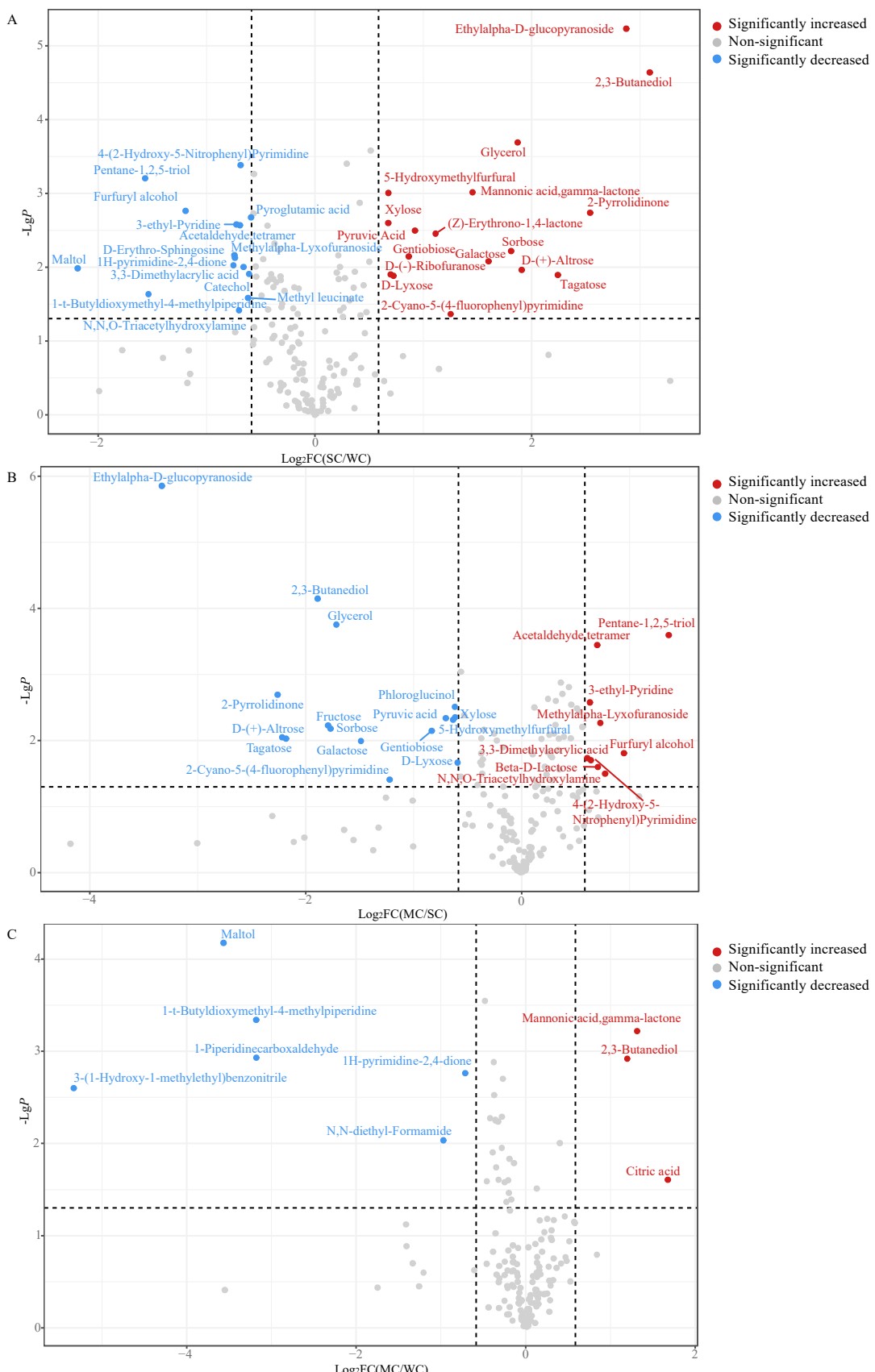

**Figure 3.** Differentially changed volatile compounds (DCVCs) between natural processing (SC), washed processing (WC), and honey processing (MC), respectively. DCVCs between SC and WC (**A**); DCVCs between MC and SC (**B**); DCVCs between MC and WC (**C**).

The compound 5-hydroxymethylfurfural showed a significant increase in SC/WC but a considerable decrease in MS/SC. This compound imparts sweet, caramel, bread, maple, brown sugar, and burnt notes to the coffee beverage [29]. Furfuryl alcohol increased in MC/SC but decreased in SC/WC, and it is associated with burnt, sweet, caramel, and brown coffee notes [33]. Therefore, the distinct sensory flavors among various treatments can be a result of the specificity of certain volatile compounds [31]. Sucrose is the main hydrocarbon present in coffee samples and serves as the primary precursor compound involved in the Maillard reaction. Moderately roasted coffee contains a significant amount of sucrose [31,32]. Isochlorogenic acid is a compound of chlorogenic acids and an important flavor-regulating substance that contributes to astringency [31,32]. It has been shown that the content of chlorogenic acids can be influenced by different drying methods [32,34].

Brazil, Colombia, Ethiopia, Honduras, Peru, Mexico, Guatemala, Nicaragua, China, and other countries are the main producers of *C. arabica*, originating from the new data of the USAD. Among these, Brazil is the largest producer of *C. arabica*, accounting for 46.41% of the world's *C. arabica* production. A total of ten furans, seven pyrroles, six pyrazines, four acids, three phenols, two pyridines, two ketones, one thiophene, one lactone, and one ether were identified in roasted *C. arabica* beans processed by different post-harvest processing methods in Brazil [35]. Although these different post-harvest processing methods did not yield characteristic volatile organic compounds, their coffee flavor still showed distinct characteristics [35]. When compared to Brazil, the volatile organic compounds in roasted *C. arabica* beans from China, particularly acids, were more abundant. An interesting finding was the similarity between DCVCs and DCnVCs, with the highest number of DCVCs observed in SC/WC followed by MS/SC and the lowest number of DCVCs found in MC/WC. This observation might be related to the fermentation process, as natural processing involves a combination of fermentation and drying, washed processing utilizes submerged fermentation, and honey processing is a hybrid of washed and natural processing [8]. Therefore, since the microbial communities developed in coffee submitted to different fermentation process vary, varying resource availability and diversity is observed [36].

### 3.3. Analysis of Sensory Characteristics

The cupping score of coffee beverages typically falls within the range of 70.00–79.00, thus indicating a premium quality (70.00–79.00) [37]. In this study, the total scores of SC, WC, and MC were 77.75, 79.50, and 77.25, respectively. The sensory evaluation included fragrance/aroma, flavor, aftertaste, acidity, body, balance, and overall impression (Figure 4), with the washed processing method receiving the highest sensory score. The sensory characteristics detected in different coffee samples subjected to different processing methods included flowery, fruity, nutty, and herbal flavors. For instance, flowery, orange fruity, and roasted walnut were described in the natural method coffee. Roasted nut, flowery, and orange were attributed to the washed method coffee, whereas roasted walnut was the dominant flavor in coffee obtained by the honey method.

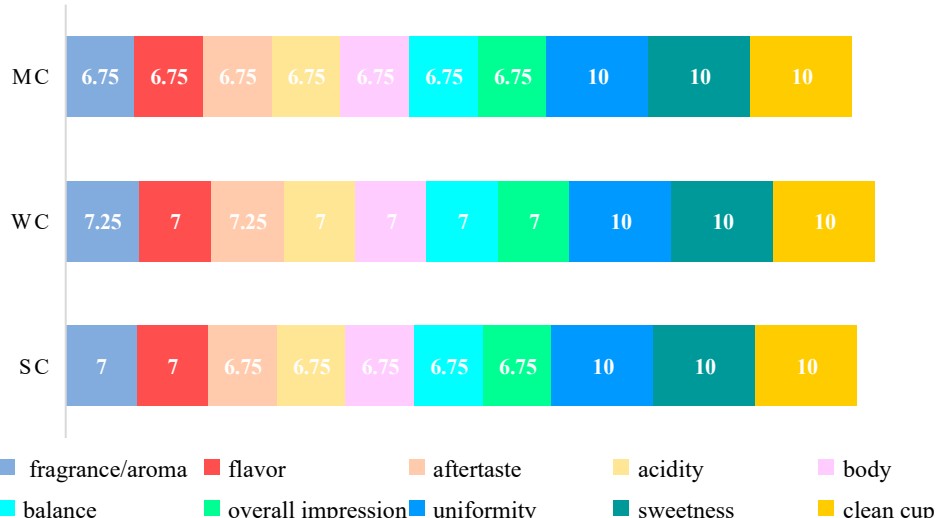

**Figure 4.** The score of the coffee cupping test.

The chemical composition of coffee strongly influences its flavor. For example, sugars showed a strong negative correlation with body, balance, aftertaste, flavor, and overall impression [32]. Compounds, such as formate, citrate, malate, 2-furyl-methanol, lipids, $\gamma$-butyro-lactone, quinic acids, acetate, and N-methyl-pyridinium were positively correlated with the body score and had a low positive correlation with balance, flavor, aftertaste, aroma and overall impression [32]. Coffee flavor formation is a complex interplay of volatile and nonvolatile compounds influenced by coffee species, geographical origins, agricultural practices, post-harvesting processing methods, roasting, brewing techniques, and storage conditions [20,24].

Natural processing results in coffee with sweet and complex body and sensory attributes because the whole cherry was dried under the sun, preventing deterioration by fungus and bacteria [38,39]. In the wet processing, the fermentation undertaken by pectinolytic microorganisms results in free amino acids, thus decreasing the contents of trigonelline, glucose, and fructose, thereby leading to high-quality coffee with less consistency and a high floral aroma [38,40]. Finally, in semi-dry processing, the mucilage is dried along with the coffee beans, imparting a honey-like or sugar-like aroma [38,41].

## 4. Conclusions

Herein, a comparative analysis of the differential nVCs and VCs, as well as the coffee flavor in roasted *C. arabica* beans from Yunnan province, was evaluated using three different primary processing methods. A total of 2642 nVCs were identified in different primary processing methods and classified into 17 classes. Moreover, 176 VCs were detected in coffee samples from different processing methods.

Furthermore, 137 DCnVCs and 32 DCVCs were detected in SC/WC. Among them, the relative levels of 39 nVCs and 15 VCs decreased significantly (VIP > 1.0, $p < 0.05$, and FC < 0.5), while 98 nVCs and 17 VCs increased significantly (VIP > 1.0, $p < 0.05$, and FC > 2). And lichenin, [6]-gingerdiol 5-acetate, 3-fluoro-2-hydroxyquinoline, and 4-(4-butyl-2,5-dioxo-3-methyl-3-phenyl-1-pyrrolidiny)benzenesulfonamide (VIP > 1.0, $p < 0.05$, and FC > 5.8 or FC < 0.1) were the important DCnVCs, and ethylalpha-D-glucopyranoside, 2,3-butanediol maltol, maltol, pentane-1,2,5-triol (VIP > 1.0, $p < 0.05$, and FC > 2.0 or FC < 0.2) were the important DCVCs.

In MC/SC, a total of 103 DCnVCs and 25 DCVCs were detected; 85 nVCs and 16 VCs decreased significantly. Meanwhile, 18 nVCs and 9 VCs increased significantly. The important DCnVCs were 3-fluoro-2-hydroxyquinoline, etimicin, lichenin, and imazamox, and the important DCVCs were ethylalapha-D-glucopyranoside, 2-pyrrolidinone, furfuryl alcohol, and pentane-1,2,5-triol. In MC/WC, 20 DCnVCs and 9 DCVCs were detected;

8 nVCs and 6 VCs decreased significantly, and 12 nVCs and 3 VCs increased significantly. The important DCnVCS were (S)-2-hydroxy-2-phenylacetonitrile O-[b-D-apiosyl-1->2)-b-D-glucoside], CMP-2-aminoethyphosphonate, talipexole, and neoconvallatoxoloside, and the important DCVCS were citric acid, mannonic acid, gamma-lactone, 3-(1-hydroxy-1-methylethyl)benzonitrile, and maltol.

Therefore, it was shown herein that the primary processing method significantly influenced the composition of coffee, as evidenced by the variation in nVCs and VCs. The optimization of coffee primary processing methods represents a promising approach to obtaining specific coffee chemical compounds and flavor profiles.

**Author Contributions:** Conceptualization, X.S. and K.L.; methodology, X.S. and K.L.; software, Y.Y., K.L. and Z.Z.; formal analysis, X.S. and C.Z.; resources, Q.W., J.S. and P.Z.; data curation, X.S. and K.L.; writing—original draft preparation, X.S.; writing—review and editing, Q.W., K.L., X.L. and J.F.; funding acquisition, X.L., C.Z. and J.F. All authors have read and agreed to the published version of the manuscript.

**Funding:** This research was funded by the Project of Yunnan Province Agricultural Basic Research Joint Foundation (No. 202101BD070001-046 and 202301BD070001-018), the National Natural Science Foundation of China-Yunnan Joint Fund (No. U1902206), and the Science and Technology Innovation Team Project of Yibin Vocational and Technical College (No. ybzy21cxtd-03).

**Institutional Review Board Statement:** Not applicable.

**Informed Consent Statement:** Not applicable.

**Data Availability Statement:** Not applicable.

**Conflicts of Interest:** The authors declare no conflict of interest.

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
