# Peer review of "Effects of Different Primary Processing Methods on the Flavor of Coffea arabica Beans by Metabolomics"

_fermentation, doi:10.3390/fermentation9080717_

Round 1
Reviewer 1 Report
The manuscript is very well written. The methodology is corect described . There are a lot of analyses presented in to the results and discussions. Conclusions are supported for the results.
The only indication to increase the quality of the manuscript is to compare with other publications in the field of research. Coffee production from China is not so well known in Europe and America comparative with countries with tradition in Coffee Arabica such as Columbia, Brazilia, Peru, Ethiopia, Nigeria, and arabic countries. A comparative study with literature will improve the quality of the paper in special the part "Differentially Changed Volatile Compounds Analysis "
Author Response
We are so grateful for your comment about our article.
It has been revised.
Reviewer 2 Report
Comments and Suggestions for Authors
The manuscript entitled “Effects of Different Primary Processing Methods on the Flavor of Coffea Arabica Beans by Metabolomics” by Xiaojing Shen et al focuses on comparing the compositions of roasted coffee beans that treated three primary processing methods including natural processing, washed processing, and honey processing, and evaluating coffee flavor characters.
This article is interesting and very detailed. I think it might be suitable for publication in this journal, however, there are some suggestions to improve the writing of this article. For instance.
Page 1, line18 “that treated three primary processing” should be “that were treated with three primary processing”
Page 1, line18 “processsing” should be “processing”
Page 2, line77 “stuy” should be “study”; “flavor for” should be “flavor to”
Page 3, lines114 and 115 “was” should be “were”
Page 5, line 222 “included” should be “including”
Page 6, line 258 “was” should be “were”
Page 6, line 270 “therefor” should be “therefore”
Page 8, line 293 “carboxyclic” should be “carboxylic”
Page 18, line 360 “flavorwere” should be “flavor were”
Page 18, line 374 “narural processing” should be “natrural processing”
Page 18, line 37 “fermentayion” should be “fermentation”
Comments and Suggestions for Authors
The manuscript entitled “Effects of Different Primary Processing Methods on the Flavor of Coffea Arabica Beans by Metabolomics” by Xiaojing Shen et al focuses on comparing the compositions of roasted coffee beans that treated three primary processing methods including natural processing, washed processing, and honey processing, and evaluating coffee flavor characters.
This article is interesting and very detailed. I think it might be suitable for publication in this journal, however, there are some suggestions to improve the writing of this article. For instance.
Page 1, line18 “that treated three primary processing” should be “that were treated with three primary processing”
Page 1, line18 “processsing” should be “processing”
Page 2, line77 “stuy” should be “study”; “flavor for” should be “flavor to”
Page 3, lines114 and 115 “was” should be “were”
Page 5, line 222 “included” should be “including”
Page 6, line 258 “was” should be “were”
Page 6, line 270 “therefor” should be “therefore”
Page 8, line 293 “carboxyclic” should be “carboxylic”
Page 18, line 360 “flavorwere” should be “flavor were”
Page 18, line 374 “narural processing” should be “natrural processing”
Page 18, line 37 “fermentayion” should be “fermentation”
Author Response
Thank you for your comments concerning our manuscript.
Those comments are all valuable and very helpful for revising and improving our paper, as well as the important guiding significance to our researches. We have studied comments carefully and have made correction which we hope meet with approval. And we checked the English writing carefully, and it was improved by editors from Mogo Internet Technology Co., LTD. We hope this improvement can be accepted.
- Comment:“Page 1, line18 “that treated three primary processing” should be “that were treated with three primary processing”
Response: This suggestion is very important and it has been revised.
- Comment:“Page 1, line18 “processsing” should be “processing”
Response: This suggestion is very important and it has been revised.
- Comment: Page 2, line77 “stuy” should be “study”; “flavor for” should be “flavor to”
Response: This suggestion is very important and it has been revised.
- Comment: Page 3, lines114 and 115 “was” should be “were”
Response: This suggestion is very important and it has been revised.
- Comment: Page 5, line 222 “included” should be “including”
Response: This suggestion is very important and it has been revised.
- Comment: Page 6, line 258 “was” should be “were”
Response: This suggestion is very important and it has been revised.
- Comment: Page 6, line 270 “therefor” should be “therefore”
Response: This suggestion is very important and it has been revised.
- Comment: Page 8, line 293 “carboxyclic” should be “carboxylic”
Response: This suggestion is very important and it has been revised.
- Comment: Page 18, line 360 “flavorwere” should be “flavor were”
Response: This suggestion is very important and it has been revised.
- Comment: Page 18, line 374 “narural processing” should be “natrural processing”
Response: This suggestion is very important and it has been revised.
- Comment: Page 18, line 377 “fermentayion” should be “fermentation”
Response: This suggestion is very important and it has been revised.
Reviewer 3 Report
The study is extensive. The method is suitable to distinguish the three primary processing methods of coffee beans. The LC/MS experimental section should include further details so that the analysis can be reproduced by others.
In Figure 1, it is not intuitive what the empty circles mean. It should be clarified.
In Figure 3, there are 2 sets of data presented via different color. A legand would be helpful to readers so that they can better understand the figure.
The manuscript should undergo a few rounds of editing for clarity. Some sentences are awkward. Here are some notable examples that need to be rewritten: line 336, "5-hydroxymethylfurfural features sweet, caramel, bread, maple, brown sugar, and burnt of coffee"; line 351, "Coffee SC, WC, and MC were premium"; line 361, "The coffee flavor was strongly concerning the chemical composition."
Author Response
Thank you for your comments concerning our manuscript. Those comments are all valuable and very helpful for revising and improving our paper, as well as the important guiding significance to our researches. We have studied comments carefully and have made correction which we hope meet with approval. And we checked the English writing carefully, and it was improved by editors from Mogo Internet Technology Co., LTD. We hope this improvement can be accepted. The main corrections in the paper and the responds to your comments are as flowing:
- Comment: “The study is extensive. The method is suitable to distinguish the three primary processing methods of coffee beans. The LC/MS experimental section should include further details so that the analysis can be reproduced by others.”
Response: We are so grateful for your comment about our article. This suggestion is very important and it has been revised.
- Comment: “In Figure 1, it is not intuitive what the empty circles mean. It should be clarified.”
Response: This suggestion is very important and it has been revised.
- Comment: “In Figure 3, there are 2 sets of data presented via different color. A legand would be helpful to readers so that they can better understand the figure.”
Response: This suggestion is very important and it has been revised.
- Comment: “The manuscript should undergo a few rounds of editing for clarity. Some sentences are awkward. Here are some notable examples that need to be rewritten: line 336, "5-hydroxymethylfurfural features sweet, caramel, bread, maple, brown sugar, and burnt of coffee"; line 351, "Coffee SC, WC, and MC were premium"; line 361, "The coffee flavor was strongly concerning the chemical composition.”
Response: This suggestion is very important and it has been revised.